

# Evaluating Uncertainty in Aerosol Forcing of Tropical Precipitation Shifts

Amy H. Peace[1], Ben B. B. Booth[2], Leighton A. Regayre[1], Ken S. Carslaw[1], David M. H. Sexton[2], Céline J. W. Bonfils[3], John W. Rostron[2]

[1]Institute for Climate and Atmospheric Science, University of Leeds, Leeds, UK
[2]Met Office Hadley Centre, Exeter, UK
[3]Lawrence Livermore National Laboratory, Livermore, CA, USA

*Correspondence to*: Amy H. Peace (cm12ap@leeds.ac.uk)

**Abstract.** An observed southward shift in tropical rainfall over land between 1950 and 1985, followed by a weaker recovery
post 1985, has been attributed to anthropogenic aerosol radiative forcing and cooling of the Northern Hemisphere relative to
the Southern Hemisphere. We might therefore expect models that have a strong historic hemispheric contrast in aerosol forcing
to simulate a further northward tropical rainfall shift in the near-term future when anthropogenic aerosol emission reductions
will predominantly warm the Northern Hemisphere. We investigate this paradigm using a perturbed parameter ensemble (PPE)
of transient coupled ocean-atmosphere climate simulations that span a range of aerosol radiative forcing comparable to multi-
model studies. In the 20[th] century, in our single-model ensemble, we find no relationship between the magnitude of pre-
industrial to 1975 inter-hemispheric anthropogenic aerosol radiative forcing and tropical precipitation shifts. Instead, tropical
precipitation shifts are associated with major volcanic eruptions and are strongly affected by internal variability. However, we
do find a relationship between the magnitude of pre-industrial to 2005 inter-hemispheric anthropogenic aerosol radiative
forcing and future tropical precipitation shifts over 2006 to 2060 under scenario RCP8.5. Our results suggest that projections
of tropical precipitation shifts will be improved by reducing aerosol radiative forcing uncertainty, but predictive gains may be
offset by temporary shifts in tropical precipitation caused by future major volcanic eruptions.

## 1 Introduction

The interaction of atmospheric aerosols with clouds and radiation is the largest cause of uncertainty in the radiative forcing of
the Earth system over the industrial period (e.g. Bellouin et al., 2020; Myhre et al., 2013). Atmospheric aerosols have short
residence times of days to weeks, therefore the strongest radiative effects of aerosols occur relatively close to emission sources.
The increase in anthropogenic aerosol emissions over the industrial era has therefore caused a negative radiative forcing mainly
in the Northern Hemisphere. This hemispheric nature of anthropogenic aerosol radiative forcing has been linked to observed
shifts in tropical precipitation and understood using idealised and transient climate model simulations (Allen et al., 2015;
Bonfils et al., 2020; Chang et al., 2011; Chemke and Dagan, 2018; Evans et al., 2020; Hwang et al., 2013; Rotstayn et al.,
2000; Rotstayn and Lohmann, 2002; Williams et al., 2001). Most notably, anthropogenic aerosol emissions that increased





rapidly across Europe and North America up to the 1980s (Lamarque et al., 2010) have been linked to an observed southward shift in tropical precipitation, which was associated with widespread drying of the Sahel between the 1950s and 1980s (Ackerley et al., 2011; Allen et al., 2015; Biasutti and Giannini, 2006; Booth et al., 2012; Dong et al., 2014; Herman et al., 2020; Hirasawa et al., 2020; Kang et al., 2021). Over recent decades there has been a partial northward return of tropical

precipitation alongside a modest recovery of rainfall over the Sahel (Giannini and Kaplan, 2019) and India, but increased drought in the Northeast Brazilian region (Utida et al., 2019). Natural aerosols from major volcanic eruptions can also cause a negative radiative forcing primarily in one hemisphere, dependent on the latitude of the eruption (Haywood et al., 2013).

Latitudinal shifts of tropical precipitation are intertwined with perturbations to the Hadley Circulation and movement of the Intertropical Convergence Zone (ITCZ). The theoretical energetic framework links the position of the ITCZ to the inter-

hemispheric energy balance. As such, a perturbation to the inter-hemispheric energy balance, particularly in the extra tropics, can trigger a shift in the position of the ITCZ and associated tropical precipitation (Kang et al., 2008, 2009, 2018a). From an atmospheric perspective, the Hadley Cell would adjust to transport more energy northwards in response to the anomalous inter-hemispheric energy balance imposed by a cooling of the Northern Hemisphere, for example, by anthropogenic aerosol forcing (Hwang et al., 2013). From the perspective of a dynamical ocean, wind-driven shallow overturning cells also act to transport

energy in the same direction as the atmosphere (Green and Marshall, 2017; Kang, 2020). Hence, in a framework where a dynamical ocean is taken into account, the atmospheric response of the ITCZ to an inter-hemispheric energy imbalance is partially dampened. In addition to aerosol radiative forcing, other forcing agents such as changes in high-latitude ice cover and ocean circulation can alter the inter-hemispheric energy balance (Broccoli et al., 2006; Chiang and Bitz, 2005; Chiang and Friedman, 2012). Migrations in tropical precipitation over the 20th century have also been linked to variability in the difference

in sea surface temperature between the Northern and Southern Hemispheres (Chiang and Friedman, 2012; Thompson et al., 2010). Because there are multiple drivers, there remains debate over whether the shifts in tropical precipitation observed over the 20th century can be attributed to anthropogenic aerosols, other forced responses, natural climate variability, or a combination of these.

Anthropogenic aerosol emissions are projected to decline in the future in response to increasingly stringent air quality and

climate change mitigation policies (Rao et al., 2017; van Vuuren et al., 2011b). Reductions in anthropogenic aerosol emissions will lead to a warming of climate relative to present day that will primarily affect the Northern Hemisphere, and could lead to a northward shift in tropical precipitation (Allen, 2015; Rotstayn et al., 2015). However, identifying the drivers of future tropical precipitation shifts is complex for a number of reasons. Warming of Northern Hemisphere landmasses caused by greenhouse gases could too lead to a northward shift in tropical precipitation (Friedman et al., 2013; Frierson and Hwang,

2012). If so, the relative warming from anthropogenic aerosol reductions and warming from increasing greenhouse gases could have an additive effect on any northward tropical precipitation shift, rather than acting in opposing directions as seen over the 20th century, making attribution to a particular forcing agent even harder to disentangle (Friedman et al., 2013). Throughout the 21st century, the climate response to warming also adds a layer of complexity to the attribution of tropical precipitation changes. Climate feedbacks, such as sea-ice feedbacks or changes in Atlantic Meridional Overturning Circulation (AMOC)



strength can modulate the position of the ITCZ, leading to a tug-of-war between different forcing and feedback components (Mamalakis et al., 2021; McFarlane and Frierson, 2017). Changes in large-scale circulation associated with warming (e.g. wet-gets-wetter paradigm) can also affect the tropical precipitation distribution and could mask an identifiable signal for latitudinal tropical precipitation shifts (Friedman et al., 2013).

Large uncertainty in anthropogenic aerosol radiative forcing limits our understanding of historical changes in climate and the
drivers of tropical precipitation and ITCZ migrations. Multi-model studies show the strength of the hemispheric difference in aerosol radiative forcing correlates with the magnitude of tropical precipitation shifts in the 20[th] century. Coupled Model Inter-comparison Project 5 (CMIP5) models with relatively detailed representations of aerosol-cloud interactions simulate a further southward migration of tropical precipitation over 1950 to 1985 (Allen et al., 2015) and better reproduce decadal drivers of Indian rainfall (Choudhury et al., 2021). However, multi-model ensembles do not sample process parameter uncertainty, so
cannot inform our understanding of which model processes influence the strength of the relationship between aerosol radiative forcing and tropical precipitation shifts. Whether or not aerosol reductions will be a main driver of tropical precipitation shifts in the future remains unclear, yet it is likely the large uncertainty in aerosol forcing will cause uncertainty in projected tropical precipitation shifts (Byrne et al., 2018a).

Here, we use a perturbed parameter ensemble (PPE) of the coupled ocean-atmosphere model HadGEM3-GC3.05 to assess the
relationship between anthropogenic aerosol radiative forcing and tropical precipitation shifts over the 20[th] and the 21[st] centuries. The PPE consists of 13 transient climate simulations with 47 parameters perturbed across a range of model schemes (Sexton et al., 2021; Yamazaki et al., 2021). By design, this ensemble spans a range of anthropogenic aerosol radiative forcing and we expect it to span a range of aerosol-driven climate responses. We compare the results from our PPE with those of earlier multi-model studies. For the 21[st] century, we compare the PPE simulations over high (RCP8.5) and low (RCP2.6)
anthropogenic emission scenarios.

## 2 Methods

### 2.1 Perturbed Parameter Ensemble of HadGEM3-GC3.05

The base model used in this work is version 3.05 of the UK Hadley Centre Unified Model (HadGEM3-GC3.05), which is a global coupled ocean-atmosphere model. HadGEM3-GC3.05 includes many of the main improvements that were made to
GC3.0 to create GC3.1 that was submitted to CMIP6 (Walters et al., 2019; Williams et al., 2018). The atmospheric component is HadGEM3-GA7.05 (Williams et al., 2018). The atmospheric and land components are configured at 'N216' resolution (approximately 60 km horizontal spacing of grid boxes at mid-latitudes), with 85 vertical atmospheric levels. HadGEM3-GC3.05 incorporates the modal version of the GLObal Version of Aerosol Processes (GLOMAP-mode) which simulates new particle formation, gas-to-gas particle transfer, aerosol coagulation, cloud processing of aerosol and aerosol deposition of
sulphate, sea salt, dust, black carbon and particulate organic species (Mann et al., 2010). The ocean and sea-ice model components are NEMO and CICE respectively (Hewitt et al., 2011). The ocean component is eddy permitting with a resolution



of ¼° and finer. Ocean components of Global Climate Models (GCMs) that are higher resolution and resolve eddies (such as this model version) can affect the mean state of the ocean, climate variability and climate response, in comparison to lower resolution ocean components (Hewitt et al., 2020).

A perturbed parameter ensemble of the above model set up was designed for UK Climate Projections 2018 (UKCP18) to sample the uncertainty in future climate changes for a given emission scenarios (Murphy et al., 2018; Yamazaki et al., 2021). This PPE samples the uncertainty in 47 model parameters from the model schemes representing convection, boundary layer, gravity wave drag, cloud radiative and microphysical properties, aerosol and land surface. The selection process for these schemes and parameters is described in detail in Sexton et al., 2021. Comprehensive filtering of the ensemble's 'parameter

space' was undertaken to identify a plausible and diverse set of model variants for generating our PPE of transient climate simulations. The filtering process first involved assessing the performance of ensemble members against observed climate variables in an atmosphere-only set up using five-day weather hind-casts and five-year simulations (Sexton et al., 2021). The model variants considered plausible after these stages were then assessed for diversity using the following idealized experiments that are similar to those used in CMIP5 protocol: aerosol effective radiative forcing (ERF) between 1860 and

2005 to 2009, ERF due to a quadrupling of $CO_2$, and sea surface temperature (SSTs) patterns prescribed for a global warming of 4 °C. Diversity was assessed using a combination of metrics from these idealized experiments, and transient coupled ocean-atmosphere simulations were created for the 25 most diverse, plausible model variants. Lastly, the transient PPE simulations were filtered based on their performance over 1900 to 2005, as described in Yamazaki et al., 2021. This multi-stage process left 15 (out of an initial 2800) remaining model variants that sample known model uncertainties and hence provide, for a given

emissions scenario, a range of climate responses. We use 13 of these ensemble members in this study, excluding a further two members on the basis that these members show steady weakening of the AMOC (Sexton et al., 2020).

Historical emissions were prescribed in the ensemble members up to 2005, then forking into the Representative Concentration Pathway (RCP) scenarios RCP8.5 and RCP2.6 to 2100. The RCPs provide a range of greenhouse gas (GHG) concentrations and emission pathways that span a range of total radiative forcing at 2100. RCP8.5 is a high emissions scenario, with GHG

emissions assumed to rise substantially out to 2100 (Riahi et al., 2011). In contrast, RCP2.6 assumes aggressive measures to substantially reduce future GHG emissions (van Vuuren et al., 2011a). The RCP scenarios assume successful implementation of air quality legislation, but RCP2.6 has approximately double the reduction of air pollutant emissions by 2030 compared to RCP8.5 (Riahi et al., 2011).

To supplement our analysis, we also use the small initial condition ensemble of four historical and SSP5-8.5 simulations

performed with the HadGEM3-GC3.1 model that were submitted to the CMIP6 archive. These simulations provide different sequences of internal variability noise for the particular emissions scenarios, and thus can be used to estimate the range of internal variability superimposed to the forced signal.





## 2.2 Quantifying Shifts in Tropical Precipitation

We define the latitudinal position of the ITCZ and tropical precipitation, $\Phi_{\text{ITCZ}}$ (degrees), as the median of annual mean
precipitation (mm day$^{-1}$) over area-weighted regional means between 20 °S and 20 °N. Several studies have used this definition
to quantify changes in the position of the ITCZ and tropical precipitation (Atwood et al., 2020; Donohoe et al., 2013; Evans et
al., 2020; Frierson and Hwang, 2012; Green et al., 2017; Green and Marshall, 2017; Moreno-Chamarro et al., 2020). We
calculate the linear trend of the 5-year rolling mean value of $\Phi_{\text{ITCZ}}$ over multi-decadal periods in the 20[th] and 21[st] centuries
over three regions – Global, Atlantic and Pacific, where the Atlantic is defined as 75 °W to 30 °E, and the Pacific as 150 °E to
75 °W.

## 2.3 Inter-Hemispheric Temperature and Radiative Forcing

To study the drivers of the ITCZ shifts, we calculate the Spearman's rank correlation coefficient of the decadal trend in $\Phi_{\text{ITCZ}}$
with variables related to the inter-hemispheric energy balance. These variables include the trend in the inter-hemispheric
difference in surface air temperature and implied total radiative forcing, plus the inter-hemispheric difference in historical
anthropogenic aerosol ERF. We also evaluate the role of cloud and non-cloud shortwave radiative responses, as well as the
role of aerosol optical depth (AOD). The inter-hemispheric difference is calculated as the difference between area-weighted
Northern Hemisphere (0 °N to 60 °N) and Southern Hemisphere (0 °S to 60 °S) means, and referred to as 'inter-hemispheric'
herein. The inter-hemispheric variable are calculates over both land and ocean, unless specified. Linear trends of the inter-
hemispheric variables are calculated from a 5-year rolling mean.
The implied total radiative forcing for the transient PPE was estimated at each grid box using the formula derived from Gregory
and Forster, 2008:

$$(1) \quad \Delta F_{\text{Im}} = \Delta F_{\text{TOA}} - \lambda \Delta T$$

where $\Delta F_{\text{Im}}$ is the implied radiative forcing of interest, $\Delta F_{\text{TOA}}$ is the change in annual mean net top of atmosphere flux relative
to a reference period, $\Delta T$ is the change in global annual mean surface air temperature relative to a reference period, and λ is
the climate feedback parameter. In this convention, positive feedback components are represented by a positive contribution
to λ. In the PPE case, the value for λ was estimated using the approach in Gregory and Forster (2008), where for an abrupt 4 x
$CO_2$ experiment, λ is the regression slope between radiative forcing and global temperature change, taking account of model
drifts in the control runs. In the case of the small HadGEM3-GC3.1 initial condition ensemble, we estimate $\Delta F_{\text{Im}}$ using a 1900
to 1920 reference period and a feedback parameter value of -0.86 W m$^{-2}$ K$^{-1}$ (following Andrews et al., 2019).
For PPE members, shortwave cloud and non-cloud radiative responses were estimated using the approximate partial radiative
perturbation (APRP) method (Taylor et al., 2007). The APRP method uses a single-layer radiative transfer model to decompose
climate model output into three components: the change in shortwave radiation due to cloud, the change in shortwave radiation
due to non-cloud atmospheric scattering and absorption, and the change in shortwave radiation due to surface albedo. Under





this method, changes in the cloud component are solely due to changes in cloud properties, whereas changes in the non-cloud
component are due to changes in aerosols, ozone and water vapour (Zelinka et al., 2014).

The idealized simulations that were used to assess the diversity of PPE members (Section 2.1; Sexton et al., (2021)), provide estimates of anthropogenic aerosol ERF between 1860 to 2005 for each PPE member in the transient coupled ocean-atmosphere simulations that we use to analyse tropical precipitation shifts. To better align with the historical time period, we completed additional simulations to provide estimates of anthropogenic aerosol ERF between 1860 and 1975 for our 13 PPE members
used in this study. ERF was quantified as the change in radiative fluxes caused by changes in anthropogenic aerosol emissions between for 1860 and 2005, plus 1860 and 1975, with SSTs, sea-ice extent and greenhouse gas concentrations held constant at 2005 to 2009 values (rather than pre-industrial values as in CMIP studies; Taylor et al., 2012).

Aerosol and physical atmosphere parameters are both important sources of uncertainty in aerosol ERF (Regayre et al., 2018). In our PPE, we perturb a total of 8 aerosol emission and process parameters in combination with multiple physical atmosphere
parameters that, amongst other responses, affect aerosol forcing. Our 13 ensemble members span a range of global mean 1860 to 2005 aerosol ERF of -2.0 to -0.9 W m$^{-2}$, which is larger and more negative, than the spread in 1850 to 2014 aerosol radiative forcing across 17 CMIP6 models (-1.37 to $-0.63$ W m$^{-2}$) (Smith et al., 2020), and similar to the estimated 1750 to 2014 aerosol ERF range from AR6 ($-2.0$ to $-0.6$ W m$^{-2}$; medium confidence) (Forster et al., 2021).

We use the inter-hemispheric 1860 to 1975 aerosol ERF when analysing tropical precipitation shifts in the 20$^{th}$ century, and
the inter-hemispheric 1860 to 2005 aerosol ERF when analysing the 21$^{st}$ century. Aerosol ERF values are shown in Figure S1-2. We do not have simulations from which to quantify the aerosol ERF in the near-term future. Hence, our analyses rely on the assumption that ensemble members with strong or weak near-present day aerosol radiative forcing will also have a strong or weak response to changes in aerosol emissions over the near-term future time periods as anthropogenic aerosol emissions decline.

**3. Results and Discussion**

**3.1 Tropical Precipitation Shifts over the 20$^{th}$ Century**

We begin by examining the latitudinal shift of tropical precipitation ($\Phi_{ITCZ}$) over the 20$^{th}$ century in our PPE. **Figure 1** shows the 5-year rolling mean evolution of $\Phi_{ITCZ}$ over the historical period in the global, Atlantic and Pacific regional means. The PPE mean $\Phi_{ITCZ}$ migrates southwards over the 1940 to 1985 period (around 0.01 °latitude year$^{-1}$ globally). This southward
migration of tropical precipitation agrees with multi-model studies (CMIP3, CMIP5) and observations of tropical precipitation over land that show tropical precipitation shifted southward in the second half of the 20$^{th}$ century (Allen et al., 2015; Bonfils et al., 2020; Chung and Soden, 2017; Hwang et al., 2013).

There are brief shifts in $\Phi_{ITCZ}$ in the years following major volcanic eruptions in the 20$^{th}$ century. The $\Phi_{ITCZ}$ time series without the 5-year smoothing applied is shown in Figure S3 to more precisely illustrate this effect. There is a northward shift in $\Phi_{ITCZ}$
following the Southern Hemisphere eruption of Mt Agung in 1963 and a southward shift following the Northern Hemisphere


eruption of El Chichón in 1982. Hence, our ensemble mean time series of $\Phi_{ITCZ}$ agrees with the literature (Bonfils et al., 2020; Colose et al., 2016; Haywood et al., 2013; Iles et al., 2013) showing the position of the ITCZ and tropical precipitation responds to volcanic eruptions and shifts away from the hemisphere with the maximum stratospheric aerosol loadings. After 1985 (the period after which pollution controls are enforced in Europe and North America), there is a northward migration of $\Phi_{ITCZ}$ to

the end of the 20$^{th}$ century. In the time series, individual members display greater inter-annual variability in the Pacific than in the Atlantic region. This could be due to different sequences of internal variability and/or different spatial-temporal evolution of the forced signal in the Atlantic and Pacific regions (e.g. Diao et al., 2021).




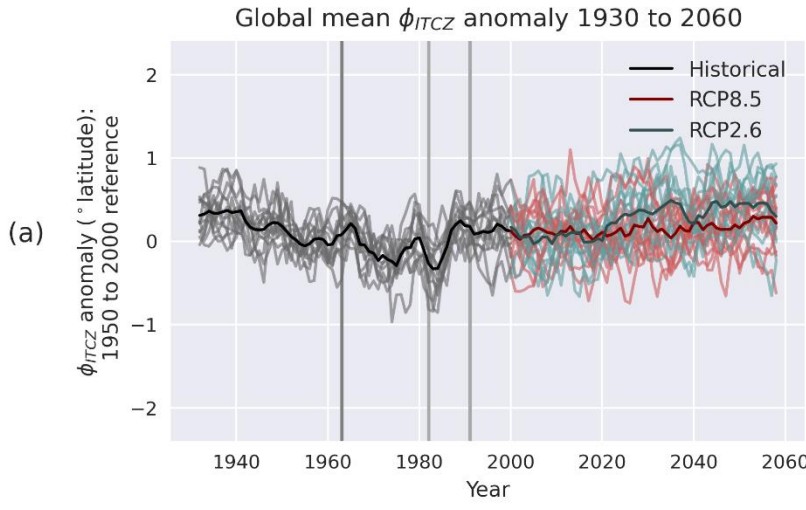

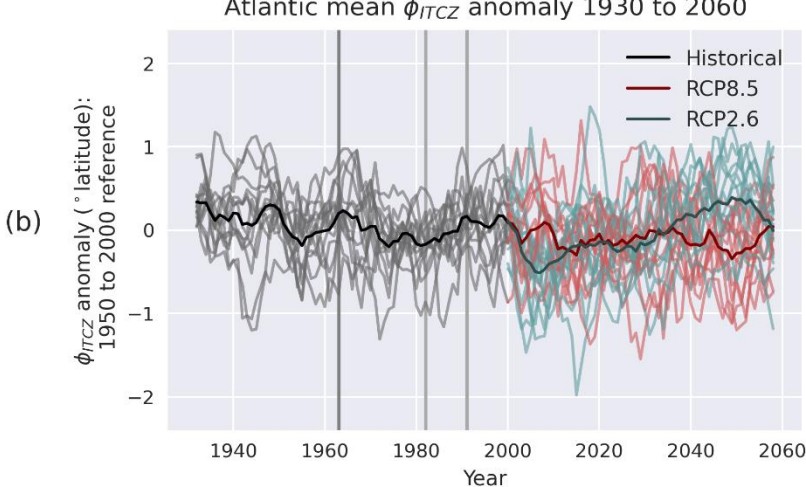

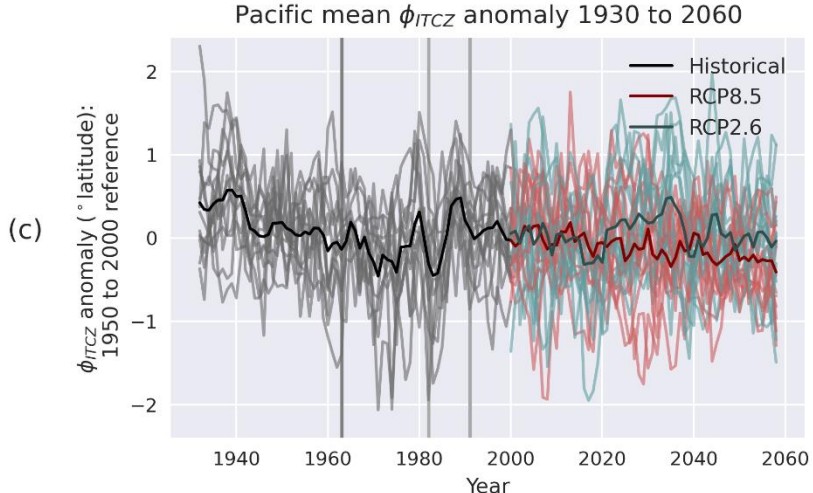

**Figure 1: Time series of 5-year rolling mean $\Phi_{ITCZ}$ anomaly relative to the 1950 to 2000 reference period for (a) global, (b) Atlantic and (c) Pacific regional means. The ensemble mean time series is shown by the darker line, and the individual ensemble members in the lighter lines. Major volcanic eruptions are marked with grey vertical lines with maximum aerosol loading in the NH (El Chichón (1982, Mexico, 17.21° N) and Pinatubo (1991, Philippines, 15.08° N)) and in the SH (Mt. Agung (1963, Indonesia, 8.20° S)).**



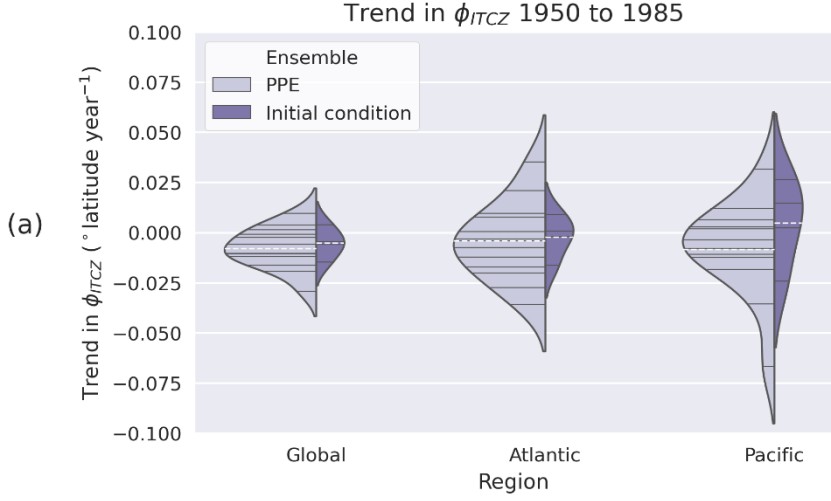

**Figure 2: Trend in 5-year rolling mean $\Phi_{ITCZ}$ for (a) 1950 to 1985 and (b,c) 2006 to 2060 for global (left), Atlantic (middle) and Pacific (right) regional means. The violin plots in light purple (in b) and red (in c) are equivalent. The black lines within the violin plots show individual ensemble members (13 in the PPE, 4 in the initial condition ensemble) and the white dashed line shows the ensemble mean.**

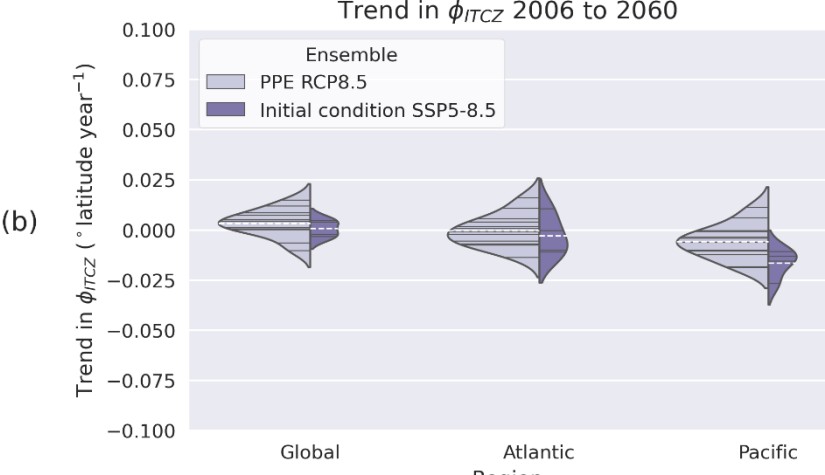

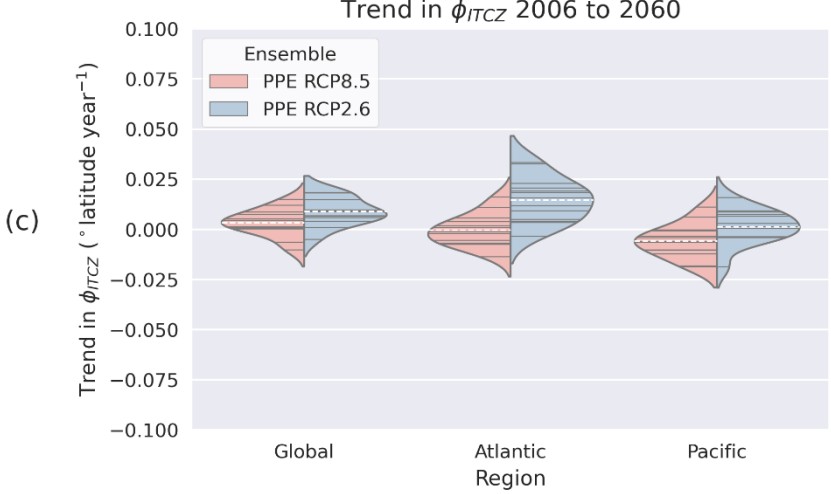





Figure 2 shows the $\Phi_{ITCZ}$ trend over 1950 to 1985 in individual ensemble members and as a density function across the PPE and initial condition ensemble. The PPE mean shows a small southward shift in the $\Phi_{ITCZ}$ throughout this time period in each region. However, across the PPE, there are both southward and northward shifts (-0.03 to 0.01 °latitude year$^{-1}$ globally). The spread in tropical precipitation shifts over 1950 to 1985 across our single-model ensemble is comparable to that over the same period from CMIP5 (see Text S1 and Figure S12) which also displayed both south and northward shifts in tropical precipitation

(Allen et al., 2015). We do not have an initial condition ensemble for each PPE member so cannot remove the effects of internal variability from each ensemble member as is common in multi-model studies. For example, in Allen et al. 2015 models that had aerosol radiative forcing experiments also had 3 to 10 ensemble members in the transient all forcing runs that were averaged to obtain the tropical precipitation shift for that model. We use the initial condition ensemble for HadGEM3-GC3.1 (a similar model version submitted to CMIP6; Andrews et al., 2020; Murphy et al., 2018) to estimate the spread in the $\Phi_{ITCZ}$

trend due to internal variability.

In the 1950 to 1985 global mean, the initial condition ensemble spread covers close to half (48%) of the spread in our PPE which suggests that a large fraction of our PPE spread is caused by natural variability, but there is still a considerable influence from perturbed parameters. Internal variability alone has been shown not to generate migrations in tropical precipitation consistent with observations over the 20$^{th}$ century (Allen et al., 2015; Chang et al., 2011). Whereas radiative forcing caused by

anthropogenic aerosol, which predominantly cooled the Northern Hemisphere, peaking in the 1980s, has been implicated as a main driver of the migration of tropical precipitation southward up to the 1980s, followed by a northward recovery to the end of the 20$^{th}$ century. We note here that models incorporating both aerosol direct and indirect effects tend to better reproduce the historical changes in temperature (Booth et al., 2012; Chung and Soden, 2017) and ITCZ location (Allen et al., 2015; Bonfils et al., 2020; Friedman et al., 2013). Our single model PPE spans a range in aerosol forcing and tropical precipitation shifts

comparable to multi-model studies. Therefore, we investigate the relationship between the uncertainty in the inter-hemispheric difference in aerosol forcing and tropical precipitation shifts in our PPE framework.

Figure 3 shows the relationships between $\Phi_{ITCZ}$ trend over 1950 to 1985 and the trend in inter-hemispheric (over 60 °S to 60 °N) surface air temperature (panel a), implied total radiative forcing (panel b) and 1860 to 1975 anthropogenic aerosol ERF (panel c). Figure S6 shows the corresponding plot but with inter-hemispheric variables calculated only over the ocean. A time

series of inter-hemispheric temperature and AOD is shown in Figure S4 and S5. There is a strong statistical relationship (r=0.91 for global mean, r = 0.66 for regional means) between the trend in inter-hemispheric surface air temperature and the $\Phi_{ITCZ}$ trend over 1950 to 1985 (with an intercept near 0). As expected, the statistical relationship between the $\Phi_{ITCZ}$ trend and the trend in inter-hemispheric implied total forcing is also strong (r >= 0.63). An energetics framework explains how the ITCZ and corresponding latitudinal position of tropical precipitation shifts in response to changes in the inter-hemispheric

distribution of energy (Kang et al., 2008, 2009, 2018b). The perturbed cross-equatorial Hadley circulation rebalances the energy asymmetry by transporting energy towards the cooler (energy deficient) hemisphere, and consequently moisture towards the warmer hemisphere. As such, Figure 3 shows that ensemble members that have greater cooling in the Northern Hemisphere and a larger difference in inter-hemispheric implied total radiative forcing over 1950 to 1985 simulate stronger





southward shifts in tropical precipitation. This behaviour is in line with the energetics theory of a southward migration of the
ITCZ due to an energy deficient Northern hemisphere. In CMIP5, models that had a stronger inter-hemispheric aerosol
radiative forcing simulated further southward shifts in tropical precipitation over 1950 to 1985, with a correlation coefficient
of r => 0.71 across 13 models (Allen et al., 2015). Despite these relationships, we do not see a strong relationship between the
strength of inter-hemispheric aerosol ERF estimated from the atmosphere-only runs and the degree of southward shift in
tropical precipitation over 1950 to 1985 in our PPE. In the paragraphs below we evaluate several hypotheses for this weaker
than expected relationship.



**Figure 3. Scatter plot of the 1950 to 1985 trend in 5-year rolling mean $\Phi_{ITCZ}$ against the 1950 to 1985 trend in inter-hemispheric (over 60 °S to 60 °N) surface air temperature (a), implied total radiative forcing (b) and anthropogenic aerosol ERF (c) for global (left), Atlantic (middle) and Pacific (right) regional means. Anthropogenic aerosol ERF is calculated over 1860 to 1975 for the PPE, and 1850 to 2014 for the initial condition ensemble. The Spearman's rank correlation coefficient between variables is shown at top left of each plot. Trend lines are shown in plots where r > 0.5.**





### 3.1.1 Potential factors obscuring a relationship between tropical precipitation shifts and anthropogenic aerosol ERF in our PPE

The bottom panel of Figure 3 shows that for the initial condition ensemble of HadGEM3-GC3.1, where model realizations have the same pre-industrial to present-day aerosol forcing, a large spread in tropical precipitation shifts is possible in transient

climate simulations due to internal variability. Although the spread in tropical precipitation shifts in our ensemble due to perturbed model parameters is larger than internal variability, this figure suggests the relationship between pre-industrial to present-day aerosol ERF and tropical precipitation shifts being obscured by internal variability in our PPE. In addition, the HadGEM3-GC3.1 initial condition ensemble explores a large fraction of the trend in the inter-hemispheric difference in temperature, which may be the reason why there is also only a weak relationship between the trend in the inter-hemispheric

difference in temperature during the 20<sup>th</sup> century and 1860 to 1975 anthropogenic aerosol ERF (Figure S7). If we had an initial condition ensemble for each PPE member or a larger sample size, the expected relationships may have emerged more strongly. The effect of internal variability is therefore likely one of the main reasons why there is not a relationship between inter-hemispheric aerosol forcing and tropical precipitation shifts over the 20<sup>th</sup> century in our ensemble.

The strength of relationships between the $\Phi_{\text{ITCZ}}$ trend and inter-hemispheric variables are also sensitive to the time period

chosen, as shown in Table 1. The 1950 to 1985 time period which has the strongest relationship between tropical precipitation shifts and both inter-hemispheric temperature and implied total forcing trends encapsulates two major volcanic eruptions. There is a weaker correlation in the longer time series or the time series excluding El Chichón. These results may indicate that volcanic rather than anthropogenic aerosol changes drive much of the coherent changes in inter-hemispheric temperature trends, implied total forcing trend, and tropical precipitation shifts during 1950 to 1985. However, the lack of a strong

relationship between tropical precipitation shifts and historical anthropogenic aerosol ERF is unlikely to be related to the choice of analysis period as there is little evidence of stronger correlations with anthropogenic aerosol ERF estimates on other 20<sup>th</sup> century timescales.

The anthropogenic aerosol ERF is quantified as the radiative change between two periods (1860 to 1975) using atmosphere-only simulations with SSTs and other climate forcers prescribed for 2005 to 2009. As such, the anthropogenic aerosol ERF

might not be representative of the aerosol radiative forcing time evolution in the transient climate simulations, due to the differences in time period, mediation of aerosol radiative effects by the coupling of ocean processes and evolution of other climate forcers, and/or internal variability (Voigt et al., 2017). To investigate further we examined the relationship between $\Phi_{\text{ITCZ}}$ trend and time evolving variables related to aerosol radiative effects, as shown in Table 2. Over 1950 to 1985, there is no clear relationship between the trend in $\Phi_{\text{ITCZ}}$ and the trend in the inter-hemispheric difference of either outgoing shortwave

radiation at TOA (Figure S8) or total AOD (Figure S9). There is some suggestion of a relationship between tropical precipitation shifts and the trend in the inter-hemispheric difference of both shortwave non-cloud radiative effect (Figure S10) and dust AOD (Figure S9), although these variables can also be affected by internal variability. Overall, the results with these time-evolving variables do not clarify well how coupling of the atmosphere to the ocean affects the relationship between





tropical precipitation shifts and the inter-hemispheric difference in anthropogenic aerosol ERF, nor explain the difference

between our results and those from a multi-model ensemble (Allen et al. 2015).

| Time period | Correlation with the trend in inter-hemispheric surface air temperature (°C year⁻¹) | Correlation with the trend in inter-hemispheric implied total forcing (W m⁻² year⁻¹) | Correlation with inter-hemispheric 1860 to 1975 anthropogenic aerosol ERF (W m⁻²) |
|---|---|---|---|
| 1950 to 1985 (shown) | r= 0.91 | r= 0.64 | r= -0.12 |
| 1950 to 1980 | r= 0.56 | r= 0.65 | r= -0.04 |
| 1940 to 1985 | r= 0.29 | r= 0.48 | r= -0.12 |
| 1940 to 1980 | r= -0.01 | r= 0.57 | r= -0.07 |
| 1940 to 1975 | r= 0.19 | r= 0.77 | r= -0.30 |

**Table 1: Table of Spearman's rank correlation coefficients for the trend in global mean 5-year rolling mean $\Phi_{ITCZ}$ and global inter-hemispheric (60 °S to 60 °N) variables shown in Figure 3 with values for alternate time periods.**




| Variable | Correlation with $\Phi_{ITCZ}$ trend (° latitude year⁻¹) |
|---|---|
| Trend in inter-hemispheric total AOD (year⁻¹) | r= -0.23 |
| Trend in inter-hemispheric dust AOD (year⁻¹) | r= -0.54 |
| Trend in inter-hemispheric shortwave non-cloud radiative effect (W m⁻² year⁻¹) | r= 0.54 |
| Trend in inter-hemispheric shortwave cloud radiative effect (W m⁻² year⁻¹) | r= -0.34 |
| Trend in inter-hemispheric top of atmosphere outgoing shortwave flux (W m⁻² year⁻¹) | r=-0.34 |


**Table 2: Table of Spearman's rank correlation coefficients for the 1950 to 1985 trend in global mean 5-year rolling mean $\Phi_{ITCZ}$ and additional global inter-hemispheric (60 °S to 60 °N) variables.**


Our 13 PPE members include the combined effects of perturbations to 8 aerosol and 39 non-aerosol parameters. So in our PPE, any relationship between anthropogenic aerosol radiative forcing and tropical precipitation might be masked by the effect of perturbations to physical atmosphere parameters. The strongest correlations between the $\Phi_{ITCZ}$ trend over 1950 to 1985 and individual perturbed parameters in our PPE are shown in Figure S11. Some of these relationships may be indicative of





important atmospheric parameter effects on tropical precipitation shifts. For example, Kang et al., 2008, 2009 showed how tuning a convective parameter related to entrainment can alter cloud radiative properties and cause a range in magnitude of ITCZ shifts for a given inter-hemispheric thermal forcing. In our PPE, the parameter that controls shallow convective core radiative effects (cca_sh_knob) and the parameter that controls the sensitivity of mid-level convection to relative humidity and entrainment (ent_fac_md) have a relationship with the $\Phi_{\text{ITCZ}}$ trend over some regions. Hence, both these parameters could modulate the sensitivity of ITCZ shifts through altering cloud radiative feedbacks. In the global and Atlantic means, the scaling of natural dimethyl sulphide emissions flux (ps_natl_dms_emiss), which could alter the hemispheric contrast of aerosol forcing, has a relationship with the $\Phi_{\text{ITCZ}}$ trend. Parameters from the land surface (related to soil moisture thresholds; psm, and altering the temperature dependence of photosynthesis; tupp_io) and the cloud microphysics (aspect ratio of ice particles; ar) scheme also have a relationship with $\Phi_{\text{ITCZ}}$ trend over some regions. These results are at best indications of possible parameter effects, since our correlations are calculated using only 13 ensemble members that conflate the uncertainty in 47 model parameters. So, further simulations would be needed to clarify parametric effects on tropical precipitation shifts.

Overall, our analysis of 20[th] century latitudinal shifts in tropical precipitation shows that any relationship between these shifts and the hemispheric contrast in 1860 to 1975 anthropogenic aerosol ERF is difficult to detect when accounting for the effect of parametric model uncertainty and internal variability. The latitudinal shift of tropical precipitation over 1950 to 1985 is, however, associated with the trend in inter-hemispheric surface temperature and implied total radiative forcing. It is also clear that major volcanic eruptions in the 20[th] century induced relatively short-lived shifts in tropical precipitation, and contribute to a time-period dependence of the strength of these relationships.

### 3.2 Tropical Precipitation Shifts Up To Mid-21[st] Century

Here, we examine the relationship between the $\Phi_{\text{ITCZ}}$ trend and our inter-hemispheric variables over 2006 to 2060 under emission scenarios RCP8.5 and RCP2.6. The reductions in anthropogenic aerosol emissions and consequential warming of the northern hemisphere in the 21[st] century have been projected to cause a northward shift in the position of ITCZ and tropical precipitation (Allen, 2015; Hwang et al., 2013). Although the warming caused by increasing $CO_2$ emissions is more homogeneous, it can also lead to a migration in the position of the ITCZ and tropical precipitation. For example, climate responses to GHG-induced warming such as ice-albedo feedbacks, the land-dominated Northern Hemisphere warming, cloud and ocean heat content changes may lead to a northward shift in the ITCZ, whereas responses such as AMOC weakening and enhanced longwave cooling may lead to a southward shift (McFarlane and Frierson, 2017). These drivers of tropical precipitation shifts in the 21[st] century will also act on different timescales. Results from multi-model studies have mixed conclusions on how zonal mean tropical precipitation will migrate in the future.

Figure 1 shows the 5-year rolling mean evolution of $\Phi_{\text{ITCZ}}$ up to 2060 across our PPE. For scenario RCP8.5, the ensemble mean $\Phi_{\text{ITCZ}}$ remains steady globally and in the Atlantic up to mid-21[st] century, with a slight southward migration in the Pacific. For RCP2.6, there is a northward migration of $\Phi_{\text{ITCZ}}$ up to mid-century globally and in the Atlantic, followed by a southward





migration from 2050 to 2060. Yet in the Pacific, the northward migration ends in around 2035 and is followed by a strong, but brief southward migration. The $\Phi_{\text{ITCZ}}$ exhibits greater variability in scenario RCP2.6, which is most pronounced in the Pacific. Figure 2 shows the $\Phi_{\text{ITCZ}}$ trend over 2006 to 2060 across individual members in our PPE and as a density function. The middle

panel of Figure 2 also shows an estimate of the impact of internal variability (superimposed on a forced signal) over the same period using the initial condition ensemble of HadGEM3-GC3.1 under emission scenario SSP5-RCP8.5. The total radiative forcing levels at 2100 are designed to be the same in RCP8.5 and SSP5-8.5 but there are differences in the emissions scenarios of individual forcing agents (Gidden et al., 2019), which could affect the evolution of tropical precipitation shifts. Under RCP8.5, the PPE mean shows only a small change in $\Phi_{\text{ITCZ}}$, with both north and southward migrations in $\Phi_{\text{ITCZ}}$ across the

ensemble. Hence, we find no robust evidence of a tropical precipitation shifts under RCP8.5 by mid-21$^{\text{st}}$ century, which is in agreement with the conclusions based on end-of-century ITCZ shifts in Byrne et al., 2018b. The spread in $\Phi_{\text{ITCZ}}$ trend due to internal variability under SSP5-RCP8.5 in the initial condition ensemble is smaller than under RCP8.5 in our PPE globally and in the Pacific, but comparable in the Atlantic region. The bottom panel of Figure 2 contrasts the $\Phi_{\text{ITCZ}}$ trend over 2006 to 2060 between RCP8.5 and RCP2.6 scenarios. By mid-century, tropical precipitation shifts further northward in RCP2.6,

compared to RCP8.5. This northward migration in tropical precipitation for RCP2.6 is in line with Allen (2015), prior to the then southward shift between mid and end of the 21$^{\text{st}}$ century. The largest difference in tropical precipitation shift between emission scenarios is in the Atlantic, which may be related to scenario dependence of AMOC strength. The AMOC is projected to weaken under warming (Collins et al., 2019; Schleussner et al., 2014), and as AMOC weakening is likely to reduce a northward ITCZ shift (McFarlane and Frierson, 2017), the effect would be more dominant in RCP8.5 than RCP2.6 due to the

greater warming from increasing GHG emissions combined with warming from anthropogenic aerosol emission reductions. In addition, the strength of the AMOC over the 20$^{\text{th}}$ century has been linked to the strength of aerosol forcing and thus aerosol forcing may also affect future projections (Hassan et al., 2021; Menary et al., 2020). Hence, the further northward migration of tropical precipitation up to 2060 in RCP2.6 in our ensemble is likely due to a combination of greater anthropogenic aerosol emission reductions in RCP2.6 compared to RCP8.5, and a greater dominance of processes in RCP8.5 that pull tropical

precipitation southwards. For example, the slight southward migration of $\Phi_{\text{ITCZ}}$ in the Pacific could be due to weakening of the Walker circulation causing, in the eastern tropical Pacific Ocean, a regional southward shift of the ITCZ (Mamalakis et al., 2021). We note here that in both emission scenarios, the spread in $\Phi_{\text{ITCZ}}$ trend across the near-term future ensembles is smaller than over the historical period, due to the trend being calculated over a longer time period. A shorter future time period induces a larger spread in the $\Phi_{\text{ITCZ}}$ trend across the PPE (Figure S14), which is comparable the spread in the historical period.

Figure 4 shows the relationship between our inter-hemispheric variables and the $\Phi_{\text{ITCZ}}$ trend over 2006 to 2060 in RCP2.6 and RCP8.5. Figure S13 shows a corresponding figure with inter-hemispheric variables calculated only over the ocean. The top panel of Figure 4 shows that there is a relationship between the $\Phi_{\text{ITCZ}}$ trend and the trend in inter-hemispheric surface temperature in each of the RCP ensembles (consistent to what we find for the historical period). For each of the RCP scenario ensembles the relationship between the global $\Phi_{\text{ITCZ}}$ trend and the trend in inter-hemispheric surface temperature is stronger

than we found in the longer historical trends (Table 1), but weaker than those beginning in 1950 which were most affected by



volcanic eruptions (Figure 3). Contrary to the historical period, we identify a relationship between the $\Phi_{ITCZ}$ trend over 2006 to 2060 and inter-hemispheric 1860 to 2005 aerosol ERF in the Pacific under RCP8.5 (r=-0.69). Emission reductions in Asia will dominate future global reductions in anthropogenic aerosol emissions (Lund et al., 2019), and align with our results that there is a strong relationship between the magnitude of inter-hemispheric aerosol ERF and tropical precipitation shifts,

particularly in the Pacific region. However, the lower latitude of emission reductions over Asia, in comparison to Europe or North America, may affect the sensitivity of the ITCZ shift. There is no relationship in the Atlantic, and consequently the global mean response (r= -0.47) is weaker than the Pacific. Figure S7 shows there is also a suggestion of a relationship between the inter-hemispheric temperature trend and 1860 to 2005 aerosol ERF in the Pacific, which is slightly stronger over the ocean (not shown). These results show that in our PPE, ensemble members that have a stronger difference in inter-hemispheric

aerosol ERF over the industrial period, and more warming in the Northern hemisphere in the near-term future under scenario RCP8.5, simulate further northward migrations in tropical precipitation, particularly in the Pacific region. It is surprising, however, that there is no clear relationship between the $\Phi_{ITCZ}$ trend and inter-hemispheric aerosol ERF in RCP2.6, where we expected faster aerosol emission reductions to yield a clearer tropical precipitation response. Possible causes of a stronger relationship between tropical precipitation shifts and aerosol radiative forcing under RCP8.5 could be due to feedbacks

between warming and aerosol radiative forcing. For example, aerosol residence times and associated net radiative effects may increase in a warmer climate (Bellouin et al., 2011; Takemura, 2020) which may amplify the effect of anthropogenic aerosol forcing on ITCZ position in RCP8.5 compared to RCP2.6.

Our analysis of 21$^{st}$ century tropical precipitation shifts suggests that the uncertainty in the inter-hemispheric difference in aerosol ERF contributes to the spread of projected tropical precipitation shifts across our ensemble in the near-term future

under RCP8.5. This is especially the case in the Pacific regional mean, as near-term future aerosol reduction will be driven by reductions from Asia. Our analysis of the historical period showed that the position of tropical precipitation can be strongly modulated by major volcanic eruptions that lead to inter-hemispheric differences in temperature. Hence, any predictive skill for future shifts in tropical precipitation will also be limited by the effect of any future major volcanic eruptions that induce differences in hemispheric energy balance.






**Figure 4: Scatter plot of trend in 5-year rolling mean** $\Phi_{ITCZ}$ **in 2006 to 2060 against the trend in inter-hemispheric (60 °S-60 °N) surface air temperature (a), and 1860 to 2005 anthropogenic aerosol ERF (b, c) for global (left), Atlantic (middle) and Pacific regional means (right). The Spearman's rank correlation coefficient is shown at top left of each plot. Trends lines are shown when r > 0.5.**



# 4. Conclusion


The inter-hemispheric nature of anthropogenic aerosol radiative forcing associated with evolving anthropogenic aerosol emissions has been linked to driving tropical precipitation shifts during the latter half of the 20th century and over the 21st century (Allen, 2015; Allen et al., 2015; Chang et al., 2011; Chemke and Dagan, 2018; Hwang et al., 2013; Rotstayn et al., 2015). In the CMIP5 multi-model ensemble there is a strong correlation between the strength of pre-industrial to present-day

inter-hemispheric aerosol forcing and the latitudinal shift in tropical precipitation over 1950 to 1985 (Allen et al., 2015). We have used a perturbed parameter ensemble of the HadGEM3-GC3.05 climate model that spans a range of aerosol forcing comparable to current generation climate models to further investigate the relationship between anthropogenic aerosol forcing and tropical precipitation shifts.

In the 20th century as anthropogenic aerosol emissions increased, our PPE mean shows a long-term southward migration in the

latitudinal position of tropical precipitation globally and in the Atlantic and Pacific up to around 1985 (e.g. 0.01 °latitude year$^{-1}$ globally over 1940 to 1985). Over the 20th century there are also brief shifts in tropical precipitation in response to major volcanic eruptions. Of the time periods we analysed, the 1950 to 1985 time period which encapsulates two major volcanic eruptions, had the strongest relationship between tropical precipitation shifts and the hemispheric contrast in temperature and implied total radiative forcing over the same period (i.e. ensemble members with more cooling in the Northern Hemisphere

simulated a further southward shift of the ITCZ). Both the long-term trends and the rapid response to volcanic eruptions are in-line with the theoretical energetic framework and modelling studies that have shown the zonal mean position of the ITCZ and corresponding tropical precipitation migrates in response to an anomalous inter-hemispheric energy balance (Kang et al., 2018b).

Despite a contemporaneous relationship between tropical precipitation shifts and the trend in the inter-hemispheric difference

in temperature and implied total forcing, we find no statistically significant relationship between the strength of inter-hemispheric 1860 to 1975 anthropogenic aerosol ERF and the magnitude of tropical precipitation shifts in the PPE over the 20th century, which contradicts results from CMIP5 (Allen et al., 2015). We propose multiple hypotheses for this different result. Overall, our results suggest that being unable to isolate forced changes from those due to internal variability (due to an absence of initial condition ensembles of our PPE members) and accounting for single-model uncertainty obscure the role of

anthropogenic aerosol forced responses in our ensemble over the 20th century.

Drivers of future tropical precipitation shifts are harder to disentangle, as both forced responses and climate feedbacks due to warming will have a bearing on the direction and magnitude of ITCZ shifts over the 21st century (McFarlane and Frierson, 2017). In the near-term future (up to 2060) globally our ensemble mean shows a negligible migration in tropical precipitation in RCP8.5, and a further northward migration in tropical precipitation in RCP2.6. The further northward migration in RCP2.6

compared to RCP8.5 is likely due to a combination of a faster reduction of anthropogenic aerosol emissions, in combination with warming-induced feedbacks (such as AMOC weakening) having a greater modulation of the regional ITCZ position in



RCP8.5. We do find ensemble members that have a stronger positive trend in inter-hemispheric temperature and forcing (i.e. due to more warming in the Northern Hemisphere) simulate further northward migrations in tropical precipitation.

In contrast to the historical time period, we find a relationship between the strength of inter-hemispheric 1860 to 2005
anthropogenic aerosol ERF (which we use as a proxy of present-day aerosol influence) and future tropical precipitation shifts under RCP8.5. This relationship is strongest in the Pacific where Asian anthropogenic aerosols have a strong historical and future influence. On the premise that present day anthropogenic aerosol influence is informative about future anthropogenic aerosol influence, this indicates ensemble members with a large hemispheric difference in historical aerosol radiative forcing, will have a further northward tropical precipitation shift in response to future aerosol reductions. Faster anthropogenic aerosol
emission reductions is likely one of the factors why RCP2.6 has a further northward shift in tropical precipitation by mid-21$^{st}$ century than RCP8.5. Yet, it is surprising that this logic does not follow through to there being a stronger relationship between tropical precipitation shifts and aerosol forcing in RCP2.6, suggesting climate feedbacks due to warming can influence the sensitivity of the climate response to aerosol forcing (Nazarenko et al., 2017; Takemura, 2020).

Overall, our study suggests the persistent uncertainty in aerosol ERF plays a role in how accurately we can project zonal mean
tropical precipitation shifts in the near-term future under RCP8.5. However, any predictive skill for future tropical precipitation shifts will also be limited by the effect of future major volcanic eruptions that can temporarily shift tropical precipitation. Our study presents open questions on the role of anthropogenic aerosol radiative forcing in modulating tropical precipitation shifts over the historical and future periods in climate models, which we cannot definitively answer here because our experiment is designed to sample single-model uncertainty and thus has a relatively small sample size and neglects the effects of internal
variability. Additional experiments that clarify the role of aerosols on near-term future tropical precipitation shifts are also needed. In a broader analysis involving multi-model and other single-model ensembles we could further develop our understanding of the relationship between feedbacks due to warming, future aerosol forcing and tropical precipitation shifts up to mid-21$^{st}$ century across multiple emission scenarios. Hence, we suggest future work investigating the role between aerosol forcing and tropical precipitation shifts in the CMIP6 multi-model ensemble and other single-model ensembles that
span a range of aerosol radiative forcing values.




**Data and code availability**

The gridded global precipitation and temperature data for the historical and RCP8.5 HadGEM3-GC3.05 PPE simulations used in this study is available in the CEDA Archive (https://catalogue.ceda.ac.uk/uuid/97bc0c622a24489aa105f5b8a8efa3f0).
Gridded global data for HadGEM3GC3.1 that was submitted to CMIP6 is available to download (https://esgf-index1.ceda.ac.uk/). The historical simulations are available under CMIP DECK and future SSP5-8.5 under ScenarioMIP. The model Source ID is HadGEM3-GC31-LL (Ridley et al., 2019a). The four members of each historical ensemble are identified by the following Variant Labels: r1i1p1f3 to r4i1p1f3. Simplified data and code required to reproduce the main figures in this article will be provided on Zonodo (link placeholder). All other underlying datasets generated during and/or analysed during
the current study are available from the Met Office Hadley Centre on reasonable request.

**Author contribution**

AP, BB, LR, KC designed the idea for this study. The analysis of data and figure preparation were completed by the AP. BB, DS and JR extracted data and performed additional aerosol forcing simulations for the HadGEM3-GC3.05 PPE. All co-authors provided discussion on the interpretation of results. AP wrote the manuscript with advice from all co-authors.

**Competing interests**

The authors declare that they have no conflict of interest.

**Acknowledgments**

AP was funded by a doctoral training grant from the Natural Environment Research Council (NERC) with a CASE studentship with the Met Office Hadley Centre under grant no. NE/P010547/1. LR and KC receiving funding from NERC under grant nos.
NE/J024252/1 (GASSP) and NE/P013406/1 (A-CURE). LR and KC received funding from the European Union's Horizon 2020 research and innovation programme under grant agreement no. 821205 (the FORCeS project). BB, DS were supported by the Met Office Hadley Centre Climate Programme funded by BEIS. JR was supported by the UK-China Research & Innovation Partnership Fund through the Met Office Climate Science for Service Partnership (CSSP) China as part of the Newton Fund. The work contribution of CB was performed under the auspices of the US Department of Energy (DOE) by
Lawrence Livermore National Laboratory under contract no. DE-AC52–07NA27344. CB was also supported by the DOE Regional and Global Model Analysis Program under the PCMDI SFA. The JASMIN facility (www.jasmin.ac.uk/) via the Centre for Environmental Data Analysis was used for data processing, which is funded by NERC and the UK Space Agency and delivered by the Science and Technology Facilities Council.



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

815    .