# Peer review of "Evaluating Uncertainty in Aerosol Forcing of Tropical Precipitation Shifts"

_Earth System Dynamics, 2022_

## Referee Comment (RC1)

Review of "Evaluating Uncertainty in Aerosol Forcing of Tropical Precipitation Shifts" by Peace et al.

General evaluation:

This research performs a large size of ensembles by perturbing 47 (or 49?) physical parameters. 13 members are selected to estimate the uncertainty of aerosol effective radiative forcing (ERF) and to investigate its relationship with ITCZ position shifts. The effects of aerosol forcing to the ITCZ positon have been explained by an energetic framework. This research provides a different result that does not support the theory previously suggested. However, this conclusion is not robust because the uncertainty of internal variability is very likely not well estimated due to small size of members (only 4 initial condition runs). In addition, some of the explanations and discussions about the PPE results are questionable. The description about experiment design is not that easy to follow. Therefore, I suggest a major revision to this manuscript.

Major comments:

1.  The description about experiment design is confusing. I have to read Sections 2 and 3 back and forth for several times, and try to guess the design. I think some more detailed explanations will help a lot. Some of the tables and figures in Supplement can be indicated more specifically in the main text to help readers follow the design.

    To my best understanding, 47 model physical parameters are perturbed. It looks like Table S1 is the list of 47 model parameters for PPE mentioned in line 102 in the main text, but Table S1 has 49 parameters instead of 47. It may be helpful to indicate Table S1 in the main text (if my guess is right) or add a paragraph to explain Table S1 in the Supplement to help readers follow the experiment design.

    Then 13 ensemble members (model variants) are selected based on their performance (i.e. diversity as mentioned in line 111). Would you please describe in more details about what is "model variant"? I guess model variant means a simulation with a set of perturbed physical parameter values that indicate the location of the model variant in the "parameter space". If my understanding is right, it is suggested to somehow rewrite this part. The concept is very abstract to me and it took me quite a long time to figure this out.

2.  In Section 2, it may be helpful to explain the concept of large ensemble simulations (or add some references). For example, how does the large set of simulations helps interpolate the "signal (the uncertainty in aerosol ERF?)" and

"noise (internal variability)". This approach is relatively new, and may not be well known to all readers at present time.

3. About the result discussion in lines 401-403, you mentioned that the relationship might be masked by perturbations to physical atmosphere parameters. The mechanisms mentioned here are meaningful *only if* the uncertainty produced by perturbing the parameter is significantly different from internal variability. I think add some discussion about the signal-to-noise analysis may help clarify the concept. I understand the member size is small and the internal variability seems not well estimated in this research (only 4 initial condition runs). Therefore, this is my main concern for the discussion about PPE.

4. The relationship between inter-hemispheric aerosol ERF and ITCZ position is bad. I think the bad relationship is probably not due to the mask effects from physical atmosphere parameters. It may be because the aerosol ERF is not the dominant factor in nature that influences inter-hemispheric energy flux (which drives Hadley circulation and influences ITCZ position). The simulations are driven by historical emission (including GHGs and anthropogenic aerosols) and fork into four different future scenarios. The aerosol ERF only accounts for a small part of the energy change, compared to GHGs. It may be easier to see the effects of aerosol ERF if the effects of GHGs can be excluded (see Wang et al. 2019).

Wang et al. (2019) Climate effects of anthropogenic aerosol forcing on tropical precipitation and circulations. *J. of Climate*. DOI: https://doi.org/10.1175/JCLI-D-18-0641.1

Another reason is the ITCZ position change belongs to a slow precipitation response (decades) through processes involving surface-atmosphere interactions. The precipitation change is shown to have small response to fast radiative processes. The details can be found in Myhre et al. 2017. The ITCZ position is mostly related to inter-hemispheric energy flux (follow Frierson's energetic framework). This is why ITCZ position is better correlated with inter-hemispheric total forcing, and worse correlated with aerosol ERF (plus, your aerosol ERF may include large internal variability). The correlation between inter-hemispheric implied total forcing and ITCZ position is not very high, but good and stable (Table 1), which also supports that this is a robust factor.

Myhre et al. (2017)  A Precipitation Driver and Response Model Intercomparison Project—Protocol and Preliminary Results, *BAMS*, DOI: https://doi.org/10.1175/BAMS-D-16-0019.1

This may also explain why the analysis using additional inter-hemispheric variables (Table 2) does not provide supporting results because all of them are fast radiative response (a few days to a few years).

5. The inter-hemispheric surface temperature also has better correlation with ITCZ position than aerosol ERF, but the relationship is very sensitive to the time period (Table 1). This is probably because the slow precipitation response must involve slow surface-atmosphere interactions/adjustments, but the processes are interrupted by volcano eruptions for a few years. The volcano eruption is a very strong impulse in aerosol amounts and can causes significant temperature change in a relatively short time period. Therefore, you see stronger correlation when the eruptions are included in the time series. Is it possible to exclude years influenced by volcano in this analysis?

Minor comments:

1. The HadGEM3-GC3.1 is perturbed by small initial conditions to estimate internal variability. Four members seems to be a very small set of ensembles. It might be better to increase the number of ensembles because the uncertainty of PPE needs to be compared with the internal variability. Or maybe you can use other methods to exclude internal variability (also see Wang et al. 2019).

2. L.165, I am not fully understanding the method quantifying ERF here.  What is the purpose/meaning of "plus 1860 and 1975"?

3. Line 348, I don't follow this sentence. What is "a large fraction of the trend…"? Can I find this information in a figure or a table?

4. You may want to do a plot the same as figure 4, but for inter-hemispheric implied total forcing, because surface temperature is too sensitive to the time period selected (as shown in historical analysis, Table 1) and may not be the best choice of variable for analyzing future projection.

Editorial suggestions:

1. Fig.S3, in the figure caption, "Historical emissions are shown in black, RCP8.5 in red and RCP2.6 in blue." is misplaced and can be eliminated.
2. Fig.S11, the title of subfigure (bottom) is "Atlantic", but it is "Pacific" in figure caption.  Please check which one is correct.
3. Add detailed indications of sub-figure in the main text may help readers to follow. For example,

line 183, global, Atlantic and Pacific → (panel a) global, (b) Atlantic and (c) Pacific
line 290, Figure 2 → Figure 2(a)
line 370 Figure S9 → Figure S9 upper panel
line 371 Figure 10S → Figure 10S upper panel
line 372 Figure S9 → Figure S9 bottom panel
line 439, Figure 2 → Figure 2(b)
… and so on.

4. The sequence of figures in Supplement is suggested to follow the sequence of appearance in the main text. For example, Figure S12 appears in line 294, Figure S6 in line 314, Figure S4 and S5 in line 315, in the main text. It is suggested to re-order figures in Supplement.

5. Lines 441 and 447, Typo, SSP5-RCP8.5 → SSP5-8.5

---

## Author Comment (AC1)

Reviewer comment= black
Reply= blue

**Review of "Evaluating Uncertainty in Aerosol Forcing of Tropical Precipitation Shifts" by Peace et al.**

**General evaluation:**

This research performs a large size of ensembles by perturbing 47 (or 49?) physical parameters. 13 members are selected to estimate the uncertainty of aerosol effective radiative forcing (ERF) and to investigate its relationship with ITCZ position shifts. The effects of aerosol forcing to the ITCZ positon have been explained by an energetic framework. This research provides a different result that does not support the theory previously suggested. However, this conclusion is not robust because the uncertainty of internal variability is very likely not well estimated due to small size of members (only 4 initial condition runs). In addition, some of the explanations and discussions about the PPE results are questionable. The description about experiment design is not that easy to follow. Therefore, I suggest a major revision to this manuscript.

We thank the referee for their comprehensive feedback. We have taken the actions detailed in blue below to improve the manuscript based on the referee's feedback.

**Major comments:**

1. The description about experiment design is confusing. I have to read Sections 2 and 3 back and forth for several times, and try to guess the design. I think some more detailed explanations will help a lot. Some of the tables and figures in Supplement can be indicated more specifically in the main text to help readers follow the design.

   This and subsequent comments by both reviewers highlight that the experimental design could be clearer. We have two responses for this.

   Firstly, the ensemble was very much an ensemble of opportunity (not one developed to address the questions in this paper), which was designed for UK Climate Projections, and we used due to a few particular properties. Hence, those who led the analysis documented in our paper were not able to influence the PPE design. The properties of the PPE and why that might have made it interesting to explore were not properly conveyed in the original submission. So, we have introduced new text into the end of the introduction that highlights this. The intention it to give the reader a conceptual overview of the PPE, and why it was used as a potential tool.

   Secondly, we have worked to improve and clarify the design process of the PPE, so that those readers who want more details can get a sense of these without reference to the existing papers that document the design.

   Specifically, the additional text clarifies the key filtering stages that were used to select the 13 PPE members used in this study from an initial pool of 2800 model parameter combinations. We have also added a schematic (below and now Figure 2) that illustrates the key stages in the design process of the PPE, and highlights in a different colour the experiments (aerosol

ERF and transient coupled ocean-atmosphere) which we use in this study to evaluate the relationship between aerosol ERF and multi-decadal tropical precipitation shifts.

[Figure]

**Figure 2: Schematic showing the stages in the design process used in UKCP18 to provide a small subset of model variants that sample a diverse climate response and are plausible when evaluated against historical climate. In this study, we use 13 PPE members of the aerosol ERF and transient coupled ocean-atmosphere experiments which are highlighted in the purple boxes.**

To my best understanding, 47 model physical parameters are perturbed. It looks like Table S1 is the list of 47 model parameters for PPE mentioned in line 102 in the main text, but Table S1 has 49 parameters instead of 47. It may be helpful to indicate Table S1 in the main text (if my guess is right) or add a paragraph to explain Table S1 in the Supplement to help readers follow the experiment design.

Thank you for pointing out this error. We were missing one parameter from the table. In the PPE, 47 independent parameters and 5 dependent parameters (that had to be perturbed to maintain consistency with independent parameters) were perturbed. The table now has 52 parameters in it (two rows have two parameters).

Table S1 containing the perturbed parameter description is referenced in the methods where we first introduce the 47 perturbed parameters. We have also lengthened the caption of Table S1 to make it more descriptive.

Then 13 ensemble members (model variants) are selected based on their performance (i.e. diversity as mentioned in line 111). Would you please describe in more details about what is "model variant"? I guess model variant means a simulation with a set of perturbed physical parameter values that indicate the location of the model variant in the "parameter space". If my understanding is right, it is suggested to somehow rewrite this part. The concept is very abstract to me and it took me quite a long time to figure this out.

Yes, that is the correct description of model variant. We agree this may be a new concept to many readers so have added a definition of model variant "Each model simulation in a PPE framework consists of a unique combination of values of the perturbed model parameters. We refer to each unique combination of parameter values as a 'model variant'" before describing the filtering process of model variants.

2.  In Section 2, it may be helpful to explain the concept of large ensemble simulations (or add some references). For example, how does the large set of simulations helps interpolate the "signal (the uncertainty in aerosol ERF?)" and "noise (internal variability)". This approach is relatively new, and may not be well known to all readers at present time.

At the end of the introduction, we have added a couple of sentences that define the advantage of using PPEs to sample parametric model uncertainty, and how that compares to multi-model that sample structural model uncertainty. In the methods section we have added that in addition to model uncertainty, uncertainty in climate projections can arise from internal variability and scenario uncertainty.

"Multi-model ensembles (MMEs) represent a collection of models which vary not only in how they represent physical processes, but in the complexity and range of processes that they represent at all. As such, it is hard to interpret differences across multi-model ensembles and link them back to processes... PPEs explore model uncertainties by perturbing influential uncertain model parameters within their plausible ranges. An advantage of using PPEs is that differences in climate response across the ensemble can often be linked back to known differences in the perturbed parameters – which can yield new insights into what causes spread in climate model response. Here, we make use of a PPE..."

"In addition to model uncertainty that can be sampled using PPEs or MMEs, uncertainty in climate projections can arise from scenario uncertainty and internal variability."

3.  About the result discussion in lines 401-403, you mentioned that the relationship might be masked by perturbations to physical atmosphere parameters. The mechanisms mentioned here are meaningful only if the uncertainty produced by perturbing the parameter is significantly different from internal variability. I think add some discussion about the signal-to-noise analysis may help clarify the concept. I understand the member size is small and the internal variability seems not well estimated in this research (only 4 initial condition runs). Therefore, this is my main concern for the discussion about PPE.

We have added the HadGEM3-GC3.1-MM (medium resolution) initial condition ensemble to our analysis that used HadGEM-GC3.1-LL (low resolution) initial condition ensemble. Each of these initial condition ensembles contains 4 members. Therefore, we have doubled our pool

of transient initial condition runs for a similar model version to better investigate the role of internal variability.

Increasing the initial condition ensemble size has affected our interpretation of how large the role of internal variability in the PPE may be. The results with the 8 initial condition ensemble members show that over 1950 to 1985 the spread in tropical precipitation shifts due to internal variability alone covers 121% and 83% of the spread in the PPE in the Atlantic and Pacific regions respectively. Hence, internal variability is likely having a large role masking an influence of aerosol radiative forcing on regional precipitation shifts in the 20[th] century and may override the parameter influence there. Our results for the global mean and 21[st] century are less affected by increasing the size of the initial condition ensemble and we still expect parameter perturbations to be having a larger influence than internal variability there.

We have added a caveat in the discussion that the uncertainty due to parametric model uncertainty may be smaller than the uncertainty caused by internal variability regionally.

"These results are at best indications of possible parameter effects. Our correlations are calculated using only 13 ensemble members that conflate the uncertainty in 47 model parameters. So, further simulations would be needed to clarify parametric effects on tropical precipitation shifts. In addition, in the Atlantic and Pacific regions the uncertainty in tropical precipitation shifts due to internal variability may be larger than parametric model uncertainty, so to robustly quantify the effects of parameters on the precipitation shifts we would need initial condition ensemble for each combination of the 47 uncertain model parameters."

4. The relationship between inter-hemispheric aerosol ERF and ITCZ position is bad. I think the bad relationship is probably not due to the mask effects from physical atmosphere parameters. It may be because the aerosol ERF is not the dominant factor in nature that influences inter-hemispheric energy flux (which drives Hadley circulation and influences ITCZ position). The simulations are driven by historical emission (including GHGs and anthropogenic aerosols) and fork into four different future scenarios. The aerosol ERF only accounts for a small part of the energy change, compared to GHGs. It may be easier to see the effects of aerosol ERF if the effects of GHGs can be excluded (see Wang et al. 2019).

Wang et al. (2019) Climate effects of anthropogenic aerosol forcing on tropical precipitation and circulations. J. of Climate. DOI: https://doi.org/10.1175/JCLI-D- 18-0641.1

We agree that the relationship between inter-hemispheric aerosol ERF and ITCZ position is bad – at least in the 20[th] century, and that an aerosol influence could be more easily isolated in an experimental set-up like Wang et al. 2019. We also expect aerosol ERF will account for a decreasing part of the inter-hemispheric energy flux into the future as anthropogenic aerosol emissions decrease. We do not have transient simulations with GHG held fixed to explore the individual contributions of forcing agents further. Creating such an ensemble in the future could help clarify the role of aerosol ERF as a cause of ITCZ shifts.

Another reason is the ITCZ position change belongs to a slow precipitation response (decades) through processes involving surface-atmosphere interactions. The precipitation change is shown to have small response to fast radiative processes. The details can be found

in Myhre et al. 2017. The ITCZ position is mostly related to inter-hemispheric energy flux (follow Frierson's energetic framework). This is why ITCZ position is better correlated with inter-hemispheric total forcing, and worse correlated with aerosol ERF (plus, your aerosol ERF may include large internal variability). The correlation between inter-hemispheric implied total forcing and ITCZ position is not very high, but good and stable (Table 1), which also supports that this is a robust factor.

Myhre et al. (2017) A Precipitation Driver and Response Model Intercomparison Project—Protocol and Preliminary Results, BAMS, DOI: https://doi.org/10.1175/BAMS-D-16-0019.1

This may also explain why the analysis using additional inter-hemispheric variables (Table 2) does not provide supporting results because all of them are fast radiative response (a few days to a few years).

This is a good point about the ITCZ shift being mainly driven by the slow precipitation response. The former version of the paper is lacking discussion on fast vs slow precipitation responses. We have added the below text to Section 3.1.1 on the representativeness of aerosol ERF and time-evolving variables related to aerosol radiative effects as a proxy for analysing slow precipitation responses.

"As such, the anthropogenic aerosol ERF might not be representative of the aerosol-driven climate response in transient climate simulations, due to the differences in time period, mediation of aerosol radiative effects by the coupling of ocean processes and evolution of other climate forcers, and/or internal variability… These results with the time-evolving variables do not clarify how representative pre-industrial to 1975 aerosol ERF is of transient aerosol radiative effects in the climate simulations. Secondly, studies have shown that the slow precipitation response to aerosol forcing is a more effective driver of ITCZ shifts than the fast precipitation response (Voigt 2017, Zhang 2021). By definition aerosol ERF does not quantity the climate response to aerosol-mediated SST changes, and hence alongside the time-evolving variables related to aerosol radiative effects, may not be a useful proxy for aerosol-driven slow precipitation changes. However, such a line of thinking does not explain the difference between our results and those from a multi-model ensemble (Allen et al. 2015)."

5.  The inter-hemispheric surface temperature also has better correlation with ITCZ position than aerosol ERF, but the relationship is very sensitive to the time period (Table 1). This is probably because the slow precipitation response must involve slow surface-atmosphere interactions/adjustments, but the processes are interrupted by volcano eruptions for a few years. The volcano eruption is a very strong impulse in aerosol amounts and can causes significant temperature change in a relatively short time period. Therefore, you see stronger correlation when the eruptions are included in the time series. Is it possible to exclude years influenced by volcano in this analysis?

The reviewer is pointing out one of the key results of our paper. As we state in the final line of the abstract "predictive gains may be offset by temporary shifts in tropical precipitation cause by future major volcanic eruptions". Some of the time period analysis shown in Table 1 has the effect of excluding the El Chichon eruption from the end of the analysis period. For example, in the 1950 to 1980 period the correlation between the trend in ΦITCZ and inter-hemispheric surface temperature trend is weaker than the 1950 to 1985 period that includes El

Chichon. We discuss the effects of volcanic eruptions on our interpretation of results extensively in section 3.

**Minor comments:**

1. The HadGEM3-GC3.1 is perturbed by small initial conditions to estimate internal variability. Four members seems to be a very small set of ensembles. It might be better to increase the number of ensembles because the uncertainty of PPE needs to be compared with the internal variability. Or maybe you can use other methods to exclude internal variability (also see Wang et al. 2019).

   We added the HadGEM3-GC3.1 medium resolution initial condition ensemble to the analysis that doubled our pool of initial condition runs (see reply to major comment 3 above).

2. L.165, I am not fully understanding the method quantifying ERF here. What is the purpose/meaning of "plus 1860 and 1975"?

   We have changed the wording of how ERF is quantified to "ERF was quantified as the change in radiative fluxes caused by changes in anthropogenic aerosol emissions between for 1860 and the industrial time period (1975 or 2005), with SSTs, sea-ice extent and greenhouse gas concentrations held constant at 2005 to 2009 values"

3. Line 348, I don't follow this sentence. What is "a large fraction of the trend..."? Can I find this information in a figure or a table?

   We have altered this sentence to be more precise. The sentence is now "the HadGEM3-GC3.1 initial condition ensembles cover nearly all (88% in the global mean) of the spread in the trend in the inter-hemispheric difference in temperature in our PPE, which may be the reason why there is also only a weak relationship between the trend in the inter-hemispheric difference in temperature during the 20th century and 1860 to 1975 anthropogenic aerosol ERF".

4. You may want to do a plot the same as figure 4, but for inter-hemispheric implied total forcing, because surface temperature is too sensitive to the time period selected (as shown in historical analysis, Table 1) and may not be the best choice of variable for analyzing future projection.

   We agree that it would a be interesting to look at the future trend in phi_ITCZ against inter-hemisphere implied total radiative forcing, and that implied radiative forcing should be a good predictor of precipitation shifts (given the historical relationships). However, the correlation coefficient between the 2006 to 2060 RCP8.5 trend in phi_ ITCZ and inter-hemispheric implied total radiative forcing is weaker than we expected. A weak relationship could be due to, for example, changing roles of forcing in future changes relative to historical (for example, could GHGs [which project onto stronger land warming] be projecting on to hemispheric contrast, mainly in continental regions which would be expected to show less influence on marine ITCZ shifts). We considered investing more analysis in understanding this further, but felt that any benefits from such insights would not change the current

inferences of the paper. We note that due to the open nature of this review that this open strand in our analysis will be there for those who may wish to follow this up.

Editorial suggestions:

1. Fig.S3, in the figure caption, "Historical emissions are shown in black, RCP8.5 in red and RCP2.6 in blue." is misplaced and can be eliminated.

   Removed from figure caption.

2. Fig.S11, the title of subfigure (bottom) is "Atlantic", but it is "Pacific" in figure caption. Please check which one is correct.

   Changed figure title to Pacific.

3. Add detailed indications of sub-figure in the main text may help readers to follow. For example,

line 183, global, Atlantic and Pacific ◊ (panel a) global, (b) Atlantic and (c) Pacific
line 290, Figure 2◊Figure 2(a)
line 370 Figure S9◊Figure S9 upper panel

line 371 Figure 10S◊Figure 10S upper panel line 372 Figure S9◊Figure S9 bottom panel line 439, Figure 2◊Figure 2(b)
... and so on.

   We have added the figure panel labels where appropriate in main text.

4. The sequence of figures in Supplement is suggested to follow the sequence of appearance in the main text. For example, Figure S12 appears in line 294, Figure S6 in line 314, Figure S4 and S5 in line 315, in the main text. It is suggested to re- order figures in Supplement.

   We have re-ordered the figures in supplement to correspond to order in main text.

5. Lines 441 and 447, Typo, SSP5-RCP8.5 ◊ SSP5-8.5

   Corrected.

---

## Author Comment (AC2)

**Reviewer comment= black**
**Reply= blue**

**Referee comment on "Evaluating Uncertainty in Aerosol Forcing of Tropical Precipitation Shifts" by Amy H. Peace et al., Earth Syst. Dynam. Discuss., https://doi.org/10.5194/esd-2022-11-RC2, 2022**

The authors use a single model ensemble to estimate the uncertainty in attributing tropical precipitation shifts to aerosols. The topic of aerosol-forced ITCZ shifts has been reported on repeatedly in the literature, but the authors here show no relationship between aerosol ERF and tropical precipitation shift, and instead argue for shifts associated with volcanic eruptions and modulated by internal variability. Following exactly what the authors did in their simulations in this paper is quite difficult. I therefore recommend major revisions.

My main comment is that the simulation description/setup is extremely difficult to follow. It seems like 13 ensemble members were eventually chosen from an initial 2800 (where does the 2800 come from?). I'm not sure at all how the 47 model parameters in the PPE map into the final 13 simulations chosen. There is a mention of a filtering process and then an assessment of diversity based on ERF from aerosols, ERF due to 4xCO2, and some other CMIP-type simulations. I feel like some kind of table or better a schematic is needed here to explicity describe what exactly the simulations are that the authors are running.

This and subsequent comments by both reviewers highlight that the experimental design could be clearer. Echoing our reply to Referee 1, we have worked to improve and clarify the design process of the PPE, so that those readers who want more details can get a sense of these without reference to the existing papers that document the design of the PPE.

Specifically, we have added more text in the methods to describe how the 13 ensemble members used in our analysis were chosen from an initial pool of 2800 model variants (parameter combinations). We have added more detailed descriptions for each of the experiments used in the filtering process, their purpose and how many model variants were retained in each stage. We have added a schematic (below) to visual this process and show which experiments form this filtering process we used in our analyses.

[Figure]

**Figure 2: Schematic showing the stages in the design process used in UKCP18 to provide a small subset of model variants that sample a diverse climate response and are plausible when evaluated against historical climate. In this study, we use 13 PPE members of the aerosol ERF and transient coupled ocean-atmosphere experiments which are highlighted in the purple boxes.**

My other main concern has to do with ensemble size. 13 member (and 4 initial condition members) do not constitute a large ensemble that can robustly estimate internal variability and uncertainty. I would like to see the authors better justify the ensemble size when work from other groups doing large ensembles (e.g. NCAR) to estimate uncertainty are using ~40 ensemble members.

We understand that the ensemble size of the PPE and initial condition ensemble is limited. However, our 13 ensemble members of the PPE were carefully selected to be both diverse and observationally plausible. Additional members with reduced plausibility would add uncertainty, but would not necessarily improve our analysis. Whereas the 4 initial condition ensemble members were those submitted to CMIP6. We have worked to address this comment in the two ways below.

Firstly, in the former version of the paper, we used the 4 members of the HadGEM3-GC3.1-LL (low resolution) initial condition ensemble to estimate the role of internal variability in tropical precipitation shifts. In the revised version, we have added the 4 members of the HadGEM3-GC3.1-MM initial condition ensemble to expand the number of ensemble members in our estimates of internal variability from 4 to 8.

Secondly, in the revised version, we have made an effort in the introduction and methods to emphasize the PPE is an 'ensemble of opportunity', in a similar fashion to the CMIP multi-model ensembles. The PPE was primarily designed to support a range of impact assessments as part of UK Climate Projections 2018, rather than address our specific research question. However, as the small sample size of the PPE has been designed to sample a broad range of climate responses, we believe it provides a unique viewpoint for assessing the relationship between aerosol forcing and tropical precipitation shifts. We have added a new figure (below and now Figure 1) that shows the PPE spans a similar range of aerosol ERF to the AR6 likely range, but a comparably smaller range of climate sensitivity. Hence, we suggest that despite the small sample size, the PPE may be a useful tool for exploring the impact of the former, rather than the latter, on ITCZ shifts, in a model with consistent physics and where differences in responses can be linked back to underlying parameters/processes.

[Figure]

**Figure 1: Range of aerosol ERF (a) and equilibrium climate sensitivity (b) across the PPE and AR6. The PPE error bars shows the 90% range across 15 PPE members compared with the AR6 90% 'very likely' ranges from (Forster et al., 2021). We use 13 PPE members in this study, excluding two shown in this figure due to model drifts.**

Finally, how generalizable are these findings? Though the authors span a range of the parameter space of certain variables, this all occurs within a single model.

The range of aerosol ERF and tropical precipitation shifts spanned within the PPE of our single model are comparable to the range spanned in AR6 and multi-model studies respectively (new Figure 1 and Text S1). Hence, we expect that in other climate models that represent similar parameters and processes, a PPE would span a similar uncertainty range. Furthermore, we compare our results to multi-model ensemble analyses in section 3 & 4 and show the benefits of evaluating uncertainty within a single model. For example, assumed relationships that emerge from multi-model analyses are not always evident when uncertainty in processes is accounted for. This suggests multi-model ensembles would benefit from a perturbed parameter component – a conclusion we think some readers will reach.